# Prognostic Factors in Patients with Sudden Cardiac Arrest and Acute Myocardial Infarction Undergoing Percutaneous Interventions with the LUCAS-2 System for Mechanical Cardiopulmonary Resuscitation

**DOI:** 10.3390/jcm11133872

**Published:** 2022-07-04

**Authors:** Michał Chyrchel, Przemysław Hałubiec, Olgerd Duchnevič, Agnieszka Łazarczyk, Michał Okarski, Rafał Januszek, Łukasz Rzeszutko, Stanisław Bartuś, Andrzej Surdacki

**Affiliations:** 1Second Department of Cardiology, Jagiellonian University Medical College, Jakubowskiego 2, 30-688 Kraków, Poland; mchyrchel@gmail.com (M.C.); rzeszutko.lukasz@gmail.com (Ł.R.); stanislaw.bartus@uj.edu.pl (S.B.); andrzej.surdacki@uj.edu.pl (A.S.); 2Student Scientific Group at the Second Department of Cardiology, Jagiellonian University Medical College, Jakubowskiego 2, 30-688 Kraków, Poland; przemyslawhalubiec@gmail.com (P.H.); olgierd.du@gmail.com (O.D.); agnieszka.lazarczyk@student.uj.edu.pl (A.Ł.); michal.okarski@student.uj.edu.pl (M.O.); 3Department of Cardiology and Cardiovascular Interventions, University Hospital in Kraków, Jakubowskiego 2, 30-688 Kraków, Poland

**Keywords:** acute myocardial infarction, angiography, cardiac arrest, LUCAS, mechanical chest compressions

## Abstract

Sudden cardiac arrest (SCA) is one of the most perilous complications of acute myocardial infarction (AMI). For years, the return of spontaneous circulation (ROSC) has had to be achieved before the patient could be treated at the catheterization laboratory, as simultaneous manual chest compression and angiography were mutually exclusive. Mechanical chest compression devices enabled simultaneous resuscitation and invasive percutaneous procedures. The aim was to characterize the poorer responders that would allow one to predict the positive outcome of such a treatment. We retrospectively analyzed the medical charts of 94 patients with SCA due to AMI, who underwent mechanical cardiopulmonary resuscitation during angiography. In total, 48 patients, 8 (17%) of which survived the event, were included in the final analysis, which revealed that 83% of the survivors had mild to moderate hyperkalemia (potassium 5.0–6.0 mmol/L), in comparison to 15% of non-survivors (*p* = 0.002). In the age- and sex-adjusted model, patients with serum potassium > 5.0 mmol/L had 4.61-times higher odds of survival until discharge from the hospital (95% CI: 1.41–15.05, *p* = 0.01). Using the highest Youden index, we identified the potassium concentration of 5.1 mmol/L to be the optimal cut-off value for prediction of survival until hospital discharge (83.3% sensitivity and 87.9% specificity). The practical implications of these findings are that patients with potassium levels between 5.0 and 6.0 mmol/L may actually benefit most from percutaneous coronary interventions with ongoing mechanical chest compressions and that they do not need immediate correction for this electrolyte abnormality.

## 1. Introduction

Sudden cardiac arrest (SCA), as a result of acute myocardial infarction (AMI) in patients qualified for urgent angiography, is a rare but severe complication (1.3% of all patients undergoing percutaneous coronary interventions (PCI)) [1]. The number of high-risk patients undergoing medical procedures at the catheterization laboratory (CathLab) is increasing and so will the occurrence of SCA in the environment of the CathLab [2]. Providing immediate cardiopulmonary resuscitation (CPR) for a patient with SCA is crucial. The CathLab, due to limited space (C-arm) and ionizing radiation, cannot be described as propitious for providing high-quality CPR. The European Resuscitation Council (ERC) Guidelines 2021 suggest that in such scenarios, mechanical chest compressions can be considered [3]. The Lund University Cardiac Assist System 2 (LUCAS-2) is one of the few available for mechanical CPR. The device provides effective chest compressions while maintaining optimal circulation [4]. Although its effectiveness does not exceed that of manual CPR [5,6], devices, such as LUCAS, have been found useful in environments not suitable for manual CPR [3,6]. Various factors have direct impact on SCA survival rate and prognosis in general, such as age, initial rhythm of electrocardiogram (ECG), delay to CPR, CPR quality and hemodynamic effect of CPR [7,8]. However, there are hardly any parameters determined as predictors of the prognosis in the scenario with mechanical CPR during treatment at the CathLab. Only a few researchers have already addressed the issue and indicated the role of aortic diastolic pressure during CPR and the length of resuscitation [9].

The aim of the current study was to find predictors of survival and evaluate their usefulness in patients qualified for urgent angiography and PCI with SCA as a consequence of AMI, treated with mechanical chest compressions.

## 2. Materials and Methods

### 2.1. Patients

We retrospectively analyzed the medical charts of patients admitted to the invasive cardiology department at one of the four centers from April 2013 to April 2020. These patients had suspected AMI. The main eligibility criterion for the final stage of analysis was that the patient had ongoing SCA resulting from AMI (before or during angiography at the CathLab), in whom the LUCAS-2 device was used for CPR.

During the investigated time period, in total, 94 patients who sustained SCA before or during angiography were treated with the LUCAS-2 device. Among 71 of them, the diagnosis of AMI was confirmed by visualization of a thrombus within the coronary artery(ies) and immediate PCI was performed. In 48 patients with AMI, data from common laboratory test results were recorded and medical history was obtained.

### 2.2. Exclusion Criteria and Ethical Approval

The following exclusion criteria were applied: age < 18 years, pregnancy, neoplastic disease, and AMI without underlying coronary artery thrombosis. The bioethics committee of our university approved the study design, accounting for the fact that the informed consent of patients (or their relatives in the case of the patient’s death) was not warranted in view of the retrospective study design (No. 1072.6120.121.2021, obtained in June 2021).

### 2.3. Mechanical Chest Compression

Essential interventions were carried out according to the Advanced Life Support (ALS) standard, including consecutive administration of vaso-constrictors and endotracheal intubation (as recommended by the ERC 2010/2015 guidelines [10]). If spontaneous return of circulation (ROSC) was not achieved within 30 min, and the patient exhibited no prognosis of improvement, resuscitation was terminated.

The LUCAS-2 device (Physio-Control Inc/Jolife AB, Lund, Sweden) provides chest compressions to a depth of 40–53 mm (set according to the subject’s sagittal plane) at a rate of 102 compressions per minute. If SCA began during angiography (IHCA), the cardiac team immediately initiated manual CPR, simultaneously preparing the LUCAS-2 device for application. The entire medical team responsible for the installation of LUCAS-2 underwent proper training. OHCA patients were subjected to manual chest compressions performed by the ambulance crew during transportation, while the LUCAS-2 device was applied immediately after arrival at the CathLab.

After successful installation, the device was immediately activated. It preserved the recommended intervals of 10 s for every 2 min of CPR in order to assess cardiac rhythm and defibrillate, if shockable. Artificial ventilation was performed with a compression-to-ventilation ratio of 30:2, continued until the patient was either intubated or a supraglottic airway device was provided, when asynchronous ventilation began. If any malfunction of LUCAS-2 was detected (e.g., battery failure), the device was removed and manual chest compressions were administered—such cases were excluded from the analysis.

### 2.4. Clinical and Biochemical Indices

Basic clinical, biochemical and laboratory data were obtained from patients’ medical records and reports from the CathLab. The primary end point was survival to hospital discharge after successful initial PCI. Additionally, being alive during transportation from the CathLab to the intensive care unit (ICU) (that is, achieved ROSC) was considered the secondary end point. We analyzed: initial cardiac rhythm, type of AMI, culprit artery, presence of coronary multi-vessel disease, initial Thrombolysis in Myocardial Infarction (TIMI) grade flow and history of accompanying diseases. Venous blood samples were collected and analyzed with the point-of-care testing (POCT) device or delivered immediately to the diagnostic laboratory. Analysis included the following: activated partial thromboplastin time (aPTT), international normalized ratio (INR), prothrombin time (PT), hemoglobin (Hb), red blood cell (RBC) and platelet count (PLT), creatinine (and estimated glomerular filtration ratio (eGFR) calculated with the formula for Chronic Kidney Disease Epidemiology Collaboration (CDK-EPI)), sodium, potassium, glucose, C-reactive protein (CRP) and the arterial blood gas test. In some subjects, the left ventricle ejection fraction (EF) from emergency echocardiography was recorded. No autopsy tests were conducted respecting the will of the patients’ families.

### 2.5. Statistical Analysis

Assuming the survival rate in the group of SCA patients undergoing coronary angioplasty with CPR by the LUCAS-2 device is close to 30% [11], the size of the group required to detect a statistically significant difference of the means for laboratory parameters was 60 patients, adopting: the power (1 − *β*) = 0.80, threshold for significance α = 0.05 and a large effect of Cohen’s group (*d*) = 0.80. If a very large group effect was observed (Cohen’s *d* = 1.30), the minimum size would be 25 patients (calculated with the T statistic and the non-centrality parameter). In the post-hoc power analysis carried out using the mean concentration of potassium in the groups (i.e., the most remarkable outcome), the calculated power of the test was equal to 0.97, and the condition for a very large Cohen group effect (*d* > 1.3) was met, which supports the validity of the above assumptions.

Continuous variables are presented as mean ± SD or median (interquartile range), as appropriate, while categorical data are presented as frequencies (*N*) and proportions (%). In the event of data loss, the case was not included in the calculations for a given outcome. Comparisons between categorical variables were performed using Fisher’s exact two-sided test. The Student’s *t*-test (or Welch’s test in the case of heterogeneity of variance by Levene’s test) was performed for mean comparisons between interval variables. If the Shapiro–Wilk test showed that the data distribution was skewed, the non-parametric Mann–Whitney U test was performed. Spearman’s correlation coefficient (*r*) was applied to determine relationships between the interval variables.

The receiver operating characteristic (ROC) curves were generated to specify the optimal cut-off values for individual laboratory parameters.

Regarding the number of outcomes (i.e., survival until discharge from hospital) being <10, univariate logistic regression was chosen to assess whether a given laboratory measure predicted the outcome. The simple multivariate model, incorporating the age and sex of patients, was investigated for each variable. A *p* value below 0.05 was considered significant. With regard to laboratory data, correction for multiple hypotheses testing was applied by performing the Benjamini–Hochberg procedure. All analyses were performed using Statistica 13.3 software (Statsoft Inc., Tulsa, OK, USA).

## 3. Results

During the period from April 2013 to September 2020, we identified 94 cases of cardiac arrest (both in and out of hospital) that required implementation of the LUCAS-2 chest compression device. Among these subjects, 71 were diagnosed with AMI and immediately transported to the CathLab. A summary regarding the available characteristics of these groups is presented in Appendix A.

The final study group of patients with AMI that received mechanical chest compressions using the LUCAS-2 system and had complete data (medical history and laboratory data) comprised 48 patients (Figure 1).

Patients were between the age of 50 and 91 (mean: 72.6 years). In 54% of patients, ST segment elevation myocardial infarction (STEMI) was noted, while in 27%, the initial rhythm was shockable (i.e., ventricular fibrillation (VF)). OHCA occurred in 75% of the patients (Table 1). There were no significant differences between patients with IHCA and OHCA (Appendix A).

Overall mortality was at 83%, leaving 8 patients in the survivor and 40 patients in the non-survivor group. Within the group of non-survivors, 33 patients died at the CathLab, and 7 at the ICU. Among patients who survived and were discharged from the hospital, the mean potassium concentration (measured during resuscitation) was markedly higher than in the patients who died (Table 2). Furthermore, the proportion of subjects with elevated serum potassium levels above 5.0 mmol/L was significantly higher among survivors (83% vs. 15%, *p* = 0.002 or *p*^BH^ = 0.02). Serum potassium concentration was correlated with serum creatinine (*r* = 0.49, *p* = 0.002 or *p*^BH^ = 0.032).

In patients with hyperkalemia, 50% underwent successful resuscitation, while the survival rate in the group with potassium < 5.0 mmol/L was equal to 4% (*p* = 0.0007). No other significant intergroup differences were identified between the survivors and non-survivors.

Univariate logistic regression analysis revealed that elevated serum potassium levels and higher creatinine were associated with higher survival rates of patients (OR: 4.60, 95% CI: 1.50–14.07, *p* = 0.008, and OR: 1.86, 95% CI: 1.02–3.41, *p* = 0.044, respectively; Table 3). After age and sex adjustment, only the serum potassium level remained statistically significant (OR: 14.18, 95% CI: 1.37–146.86, *p* = 0.03; Table 3).

This relationship was also significant when potassium levels were treated as a dichotomous variable. The aforementioned analysis yielded hyperkalemia as the only predictor of survival, the odds being almost 10-times higher when compared to patients with potassium levels below 5.0 mmol/L (OR: 9.8, 95% CI: 1.6–60.5, *p* = 0.01; in the age- and sex-adjusted model: OR: 4.61, 95% CI: 1.41–15.05, *p* = 0.01; Appendix A.).

No associations were detected between survival and other laboratory parameters, or with data regarding the angioplasty procedure. Substantially, no significant relationship between potassium concentration and pH or glycemia was found.

As the mild to moderately elevated serum potassium level was determined to be associated with a higher survival rate in the investigated group, it was decided to generate a ROC curve with serum potassium concentration as a predictor and survival rate as an outcome (Figure 2).

The best cut-off point was determined using the highest Youden’s index that yielded optimal balance between sensitivity and specificity. Therefore, the highest Youden’s index of 0.71 was observed for a cut-off value of 5.1 mmol/L, resulting in 83.3% sensitivity and 87.9% specificity, suggesting that the choice of 5.0 mmol/L as a criterion for hyperkalemia was, in this case, justified. The area under the ROC curve for all-cause deaths was 0.87 (95% CI = 0.72–1.00, *p* < 0.001).

ROC curve analysis with the tangent method allowed us to determine that the optimal cut-off value for serum potassium to predict survival was 5.4 mmol/L, with a sensitivity of 66.7% and a specificity of 93.9%.

## 4. Discussion

The main finding in the current study was that an increased potassium level (>5.0 mmol/L) was observed in patients who survived resuscitation at the CathLab. In the group of 48 patients subjected to detailed analysis, ROSC was obtained for 31%, while in the group of 71 patients diagnosed with AMI, 34% obtained ROSC. The proportion of those who survived until discharge from the hospital was 17% and 24%, respectively.

In the study by Hardig et al., ROSC was obtained in 51% of patients who underwent resuscitation at the CathLab using mechanical chest compression devices, while 26% survived until hospital discharge. Multiple characteristics of patients investigated in that study were comparable to those achieved in this work. Examples include the age of patients (median of 72 years), initial ECG rhythm (shockable in 20%) and comorbidities (HT 51%, DM2 20% and chronic kidney disease (CKD) 6%). Only STEMI was almost 7-times more frequent than NSTEMI [9], while in our research, it was 2-times more prevalent.

Wagner et al. observed that ROSC occurred in 37% of patients, while hospital discharge was possible in 28%. The median age (73 years), the incidence of comorbidities and the initial rhythm of SCA were close to those perceived in our patients. The main difference was also the proportion of STEMI, equal to 77%; however, it did not markedly influence overall survival compared to our analysis [12].

The distinctive finding of this research was that mild hyperkaliemia predicted a positive outcome of LUCAS-2-assisted resuscitation at the CathLab.

We proposed a cut-off point of 5.0 mmol/L [11] in our calculations because it is closer to the highest Youden index value obtained for a potassium concentration of 5.1 mmol/L. It is noteworthy that choosing the value of 5.5 mmol/L (frequently used in the literature on the subject [13]) did not qualitatively affect the interpretation of results in this study.

In the analysis conducted by Hardig et al., it was the time from SCA recognition to the beginning of CPR that determined ROSC and survival. In fact, only 22% of the patients with OHCA had ROSC, while none of them survived to discharge from the unit [9]. Nonetheless, the most crucial factor having impact on these results is the time from SCA to the introduction of life support procedures, and then, transportation to the hemodynamics ward, which was not provided in that study. Analogous results were obtained by Almalla et al. In their study, from 56 patients with OHCA that underwent PCI during CPR with LUCAS, only 1 survived to hospital discharge [14]. Similarly, in the case series described by Wagner, only the subjects with SCA at the CathLab survived, but not the patients with OHCA [12].

In our study, the higher survival rate in this group could be partially explained by the fact that two survivors with OHCA had intra-aortic balloon pumps (IABP), and one underwent therapeutic hyperthermia. Another two of these patients had ROSC before arriving at the CathLab and then, a second SCA during angiography.

Interestingly, from Table 2, a significant difference in CRP can be observed between the groups; however, the “markedly elevated CRP (326 mg/dL)” was noted in only one patient, thus, skewing the data interpretation. In order to compare the survivor group with the non-survivor group, we performed nonparametric tests that are known to be robust to outliers, which revealed that even after taking such data points into account, there was no significant difference between the groups.

The fact that potassium concentration was positively correlated with kidney function (i.e., eGFR or serum creatinine) is of great importance. As the detailed medical history of some patients might not have been complete, we could simply hypothesize, based on laboratory data, that the observed hyperkaliemia was caused by CKD, and, thus, it was a chronic abnormality. This phenomenon could have led to adaptation to slightly elevated potassium concentrations, which may be crucial for understanding how mild hyperkalemia explains higher survival. Einhorn et al. showed that the risk of death during hyperkalemia decreases two-fold in the presence of CKD (stages ≥ 3) [15].

A possible mechanism of the aforementioned phenomenon is the protective effect of higher potassium concentrations on the heart muscle, analogous to the action of the cardioplegic solution. In a model incorporating neonatal rat cardiomyocytes subjected to ischemia, hyperkalemia delayed and inhibited the increase in diastolic [Ca^2+^]. Prevention of calcium accumulation in cardiomyocytes suggests a protective nature during ischemic state [16]. The supposed cardioprotective effect of potassium was also considered in an experiment carried out by Lee et al. After induction of SCA in 20 pigs, they waited 14 min, and then with the commencement of CPR, half of the animals had a dose of 0.9 mEq/kg of KCl diluted to 20 mL administered into the right atrium of the heart (the rest were given a placebo). ROSC was achieved in 70% of the animals supplied with potassium and only 20% of those receiving placebo. This, again, suggests that higher potassium levels might act beneficially on the heart during SCA [17]. This mechanism seems even more likely as some other factors, such as pH or glycemia—generally considered as critical in patients with RCA—were neither associated with prognosis nor correlated with potassium concentration in a significant manner.

There is some indirect evidence that, as pointed out by Einhorn et al. [15], in the population with CKD, chronic hyperkalemia can affect the membrane potential of cardiomyocytes. This would result in an increased capability to compensate the additional electrolytic and metabolic disturbances as those observed during an ischemic episode.

It appears that the protective mechanism of mild hyperkalemia in myocardial cells during SCA could stem from the adaptation of myocardial cells to higher potassium concentrations. In CKD, there are numerous ion disturbances that, by affecting the membrane potential, reduce the sensitivity of cells to interstitial hyperkalemia. Since CPR conduced with LUCAS-2 provides up to 60% of physiological blood flow through the circulatory system [12], it might be enough to enable ROSC in some subjects.

The obtained results allow one to indicate that about one in five patients with SCA who were treated at the CathLab with LUCAS-2-performed CPR could survive the event, and this result is consistent with the literature [9,12]. We cannot conclude whether mechanical chest compressions were the factor that determined survival, as other approaches were described. For example, in the study described by Almalla et al. [14], the majority of patients were resuscitated until ROSC and then transferred to the CathLab. What can be stated at this point is that LUCAS-2 allowed recanalization of coronary vessels during CPR, which was crucial to eliminate the initial cause of SCA.

Furthermore, our observations suggest that in these patients, mild hyperkalemia (5.0–6.0 mmol/L) might not require immediate correction and may perhaps even play a protective role. Such outcomes have already been anticipated in the literature [18]. These conclusions almost certainly do not apply to potassium concentrations >6.0 mmol/L, since this level poses a life-threatening condition.

Thus, at the moment, we perceive mild hyperkalemia primarily as a biomarker of better prognosis in patients with SCA caused by AMI who undergo CPR and simultaneous PCI. Hyperkalemia can be easily detected using the POCT system. We propose a strict cut-off value of potassium concentration at 5.0 mmol/L (in concordance with common definition of mild hyperkalemia, reasonably close to the value calculated from the ROC) that could, for example, facilitate decisions regarding extended resuscitation.

Our study has some limitations, including the relatively small group of examined patients (however, it possesses adequate statistical power to draw the discussed conclusions) and significant data loss, which lessen our ability to make some inferences. Our study is a retrospective study that may be an inspiration for large-scale quantitative research. It contributes to the creation of new theses, the confirmation of which in large randomized prospective studies may become the basis for drawing conclusions that can then be used in clinical practice.

## 5. Conclusions

Mechanical chest compressions with the LUCAS-2 device allowed successful resuscitation in selected patients. Mild to moderate hyperkalemia during mechanical cardiopulmonary resuscitation performed at the CathLab was found to be a predictor of greater survival rate. The hypothetical consequences of our discoveries for clinicians are that in patients with potassium levels of 5.0–6.0 mmol/L, the decision to cease the resuscitation could be delayed. Moreover, immediate correction of this electrolyte abnormality might not be warranted. Prospective research is required to provide confirmation of these initial findings.

## Figures and Tables

**Figure 1 jcm-11-03872-f001:**
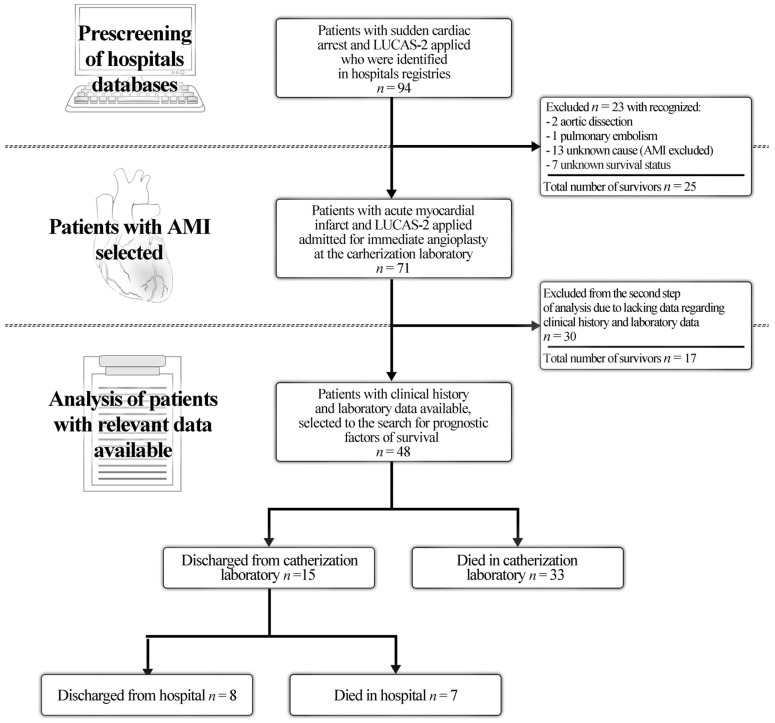
The study flowchart. The particular steps of data acquisition and analysis are shown. AMI: acute myocardial infarction. AMI: Acute Myocardial Infarction; LUCAS-2 Lund University Cardiac Assist System 2.

**Figure 2 jcm-11-03872-f002:**
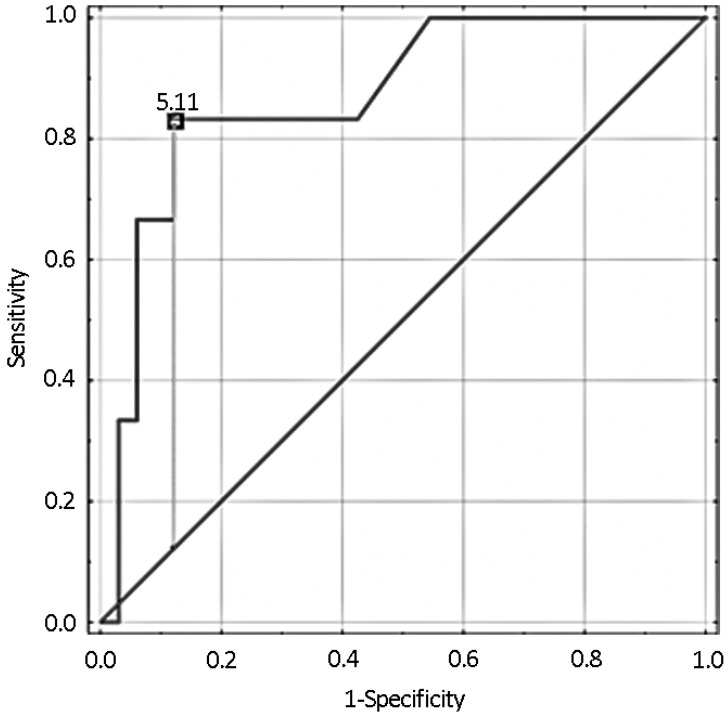
Receiver operating characteristic curve (ROC) representing efficacy of potassium concentration measured during resuscitation in prediction of survival until hospital discharge.

**Table 1 jcm-11-03872-t001:** Characteristics of patients and outcomes.

**Selected Indices**	**Overall Group** ***n* = 48 (100%)**	**Survivors** ***n* = 8 (16.7%)**	**Non-Survivors** ***n* = 40 (83.3%)**	***p*-Value**
Age, years	72.6 ± 11.37	71.4 ± 177	72.8 ± 10.9	0.9
Gender, males	27 (56%)	6 (75%)	21 (53%)	0.4
LVEF, (N, (%))	26.8 ± 15.3	17.5 ± 14.8	28.0 ± 15.5	0.4
**Myocardial Infarction Type**				
NSTEMI/STEMI/unknown, (N, (%))	12 (25)/26 (54)/10 (21)	1 (13)/3 (38)/4 (49)	11 (28)/23 (58)/6 (14)	1.0
**Onset of Cardiac Arrest**				
OHCA/IHCA ^a^, (N, (%))	25 (52)/23 (48)	6 (75)/2 (25)	19 (48)/21 (52)	0.2
**Initial Rhythm at Cardiac Arrest**				
VF/PEA/asystole/unknown, (N, (%))	13 (27)/22 (46)/ 8 (17)/5 (10)	1 (13)/5 (63)/ 0/2 (24)	12 (30)/17 (43)/ 8 (20)/3 (7)	0.2
**Patient History**				
Hypertension, (N, (%))	28 (58)	3 (38)	25 (63)	0.1
Type 2 Diabetes, (N, (%))	12 (25)	2 (25)	10 (25)	1.0
Prior myocardial infarction, (N, (%))	10 (21)	1 (13)	9 (23)	1.0
Arrhythmias ^b^, (N, (%))	11 (23)	1 (13)	10 (25)	1.0
Valvular heart disease, %	4 (8)	1 (13)	3 (8)	0.5
Chronic kidney disease, (N, (%))	5 (10)	1 (13)	4 (10)	0.6
ICD, (N, (%))	3 (6)	0 (0)	3 (8)	1.0
**Coronary Angiography**				
**Distribution of Diseased Vessels**				
LM/LAD/Cx/RCA, (N, (%))	13 (27)/36 (75)/ 23 (48)/27 (56)	1 (13)/7 (88)/ 4 (50)/5 (63)	12 (30)/29 (73)/ 19 (48)/22 (55)	-
**Infarct-Related Arteries**				
LM/LAD/Cx/RCA, (N, (%))	10 (21)/22 (46)/ 10 (21)/14 (29)	1 (13)/1 (13)/ 1 (13)/3 (38)	9 (23)/21 (53)/ 9 (23)/11 (28)	-
**TIMI Grade Flow at Baseline**				
TIMI LM > 1, (N, (%))	29 (60)	4 (50)	25 (63)	1.0
TIMI LAD > 1, (N, (%))	16 (33)	4 (50)	12 (30)	0.4
TIMI Cx > 1, (N, (%))	24 (50)	3 (38)	21 (53)	0.7
TIMI RCA > 1, (N, (%))	19 (40)	2 (25)	17 (43)	1.0
Multivessel CAD, (N, (%))	15 (31)	1 (13)	14 (35)	0.6
2 vessels, (N, (%))	11 (23)	1 (13)	10 (25)	1.0
3 vessels, (N, (%))	4 (8)	0 (0)	4 (10)	1.0
**Procedural Data**				
Volume of contrast, mL	150 [135–260]	150 [150–250]	150 [90–320]	0.7
Radiation dose, mGy	1131 ± 1150	582 ± 237	1240 ± 1233	0.2
Respirator, (N, (%))	41 (85)	7 (88)	34 (85)	1.0
PCI, (N, (%))	45 (94)	7 (88)	38 (95)	1.0
Radial/femoral/unknown PCI access, (N, (%))	10 (21)/33 (69)/5 (10)	1 (12.5)/6 (75)/ 1 (12.5)	9 (23)/27 (68)/4 (9)	1.0
Balloon angioplasty, (N, (%))	39 (81)	7 (88)	33 (83)	1.0
Stent, (N, (%))	36 (75)	7 (88)	29 (73)	0.6
Thrombectomy, (N, (%))	13 (27)	3 (38)	10 (25)	0.4
Gp IIb/IIIa inhibitors, (N, (%))	18 (38)	3 (38)	15 (38)	1.0
Endocavitary electrode, (N, (%))	16 (33)	1 (13)	15 (38)	0.4
IABP, (N, (%))	9 (19)	2 (25)	7 (18)	0.6
Pressor amines				
Adrenaline, (N, (%))	21 (44)	4 (50)	17 (43)	0.7
Noradrenaline, (N, (%))	22 (46)	4 (50)	18 (45)	0.7
Dobutamine, (N, (%))	18 (38)	3 (38)	15 (38)	0.7
Dopamine, (N, (%))	13 (27)	2 (25)	11 (27)	0.7

Data are shown as mean ± standard deviation (SD) or nominal values and percentages (%). *p*-values below 0.05 are marked in bold. ^a^ OHCA/IHCA refers to the rhythm observed at the initial contact with patients with SCA. However, 4 patients with OHCA had ROSC before the LUCAS was mounted and then, suffered the next episode of SCA. Of these patients, 2 survived until discharge from hospital. ^b^ Arrythmias include: atrial fibrillation, 3rd-degree cardiac block and left bundle branch block. LVEF: left ventricular ejection fraction; NSTEMI: non-ST segment elevation myocardial infarction; STEMI: ST segment elevation myocardial infarction; OHCA: out-of-hospital cardiac arrest; VF: ventricular fibrillation; PEA: pulseless electrical activity; ICD: implantable cardioverter defibrillator; LM: left main coronary artery; LAD: left anterior descending coronary artery; Cx: circumflex coronary artery; RCA: right coronary artery; TIMI: thrombolysis in myocardial ischemia; CAD: coronary artery disease; PCI: percutaneous coronary intervention; IABP: intra-aortic balloon pump.

**Table 2 jcm-11-03872-t002:** Biochemical indices.

Selected Indices	Overall Group *n* = 48 (100%)	Survivors *n* = 8 (16.7%)	Non-Survivors *n* = 40 (83.3%)	*p*-Value
Prothrombin time, s	19 [12.7–16.9]	13.5 [13.9–25.8]	14.4 [12.0–15.4]	0.06
Hemoglobin, g/dL	12.4 ± 2.3	12.7 ± 2.9	12.3 ± 2.2	0.7
Platelets, ×10^3^/μL	193.7 ± 75.6	163.7 ± 70.6	200.3 ± 76.1	0.3
Creatinine, mg/dL	1.8 ± 1.2	2.7 ± 1.5	1.6 ± 1.1	0.07
Sodium, mmol/L	138.6 ± 4.9	139 ± 5	139 ± 5	0.7
Potassium, mmol/L	4.2 [4.0–5.0]	5.8 [5.1–6.4]	4.2 [3.9–4.7]	0.003 ^a^
Glucose, mg/dL	196 [128.0–302.4]	284 [162.0–406.8]	196 [127.8–302.4]	0.5
CRP, mg/dL	5.9 [4.6–21.2]	64.8 [1.0–326.4]	5.5 [4.6–12.0]	0.5
pH	7.15 ± 0.20	7.19 ± 0.18	7.13 ± 0.21	0.5
pCO_2_, mmHg	46.3 ± 16.3	40 ± 18	48 ± 16	0.4
pO_2_, mmHg	68 [46–110]	79.3 [78.8–112.0]	67.7 [38.0–101.2]	0.2

Laboratory data. ^a^
*p* = 0.03 after the correction for multiple comparisons by means of the Benjamini–Hochberg procedure. CRP: C-reactive protein.

**Table 3 jcm-11-03872-t003:** Univariate logistic regression analysis of the predictors of survival with laboratory parameters as continuous variables.

Predictor	Odds Ratio	95% Confidence Interval	*p*-Value
Age, per 10 years increment	0.90	0.45–1.81	0.8
Sex, male vs. female	1.65	0.70–3.89	0.3
Prothrombin time, per 1 s increment	1.32	0.92–1.89	0.1
Hemoglobin, per 1 g/dL increment	1.09	0.76–1.57	0.6
Platelets, per 50 × 10^3^/μL increment	0.71	0.40–1.27	0.3
Creatinine, per 1 mg/dL increment ^a^	1.81	0.97–3.37	0.06
Sodium, per 10 mmol/L increment	1.05	0.17–6.38	1.0
Potassium, per 1 mmol/L increment	14.18	1.37–146.86	0.03
Glucose, per 10 mg/dL increment	1.05	0.92–1.21	0.5
CRP, per 10 mg/dL increment	1.17	0.93–1.48	0.2
pH, per 1 unit increment	1.08	0.84–1.39	0.5
pCO_2_, per 10 mmHg increment	0.70	0.36–1.38	0.3
pO_2_, per 10 mmHg increment	1.11	0.93–1.32	0.3

^a^ Age- and sex-adjusted ORs and *p*-values.

## Data Availability

The data may be made available for a justified request.

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
