# Peer review of "Prognostic Factors in Patients with Sudden Cardiac Arrest and Acute Myocardial Infarction Undergoing Percutaneous Interventions with the LUCAS-2 System for Mechanical Cardiopulmonary Resuscitation"

_jcm, 2022, doi:10.3390/jcm11133872_

Round 1
Reviewer 1 Report
Chyrchel et al present a small retrospective study on prognostic factors in patients with sudden cardiac arrest and 2 acute myocardial infarction undergoing percutaneous 3 interventions with the LUCAS-2 system for mechanical 4 cardiopulmonary resuscitation, with the main finding being those with serum potassium >5.0 mmol/L had 4.61 times higher odds of survival until discharge from the hospital. This is a retrospective study and the results should be interpreted very carefully.
- In discussing potassium levels in an arrest case, we have to emphasize and discuss pH. Ineffective CPR or delayed CPR is associated with low pH, and survival is dismal with lower pH. Why is this different here? Are the findings independent of pH? What was lactate level in each group? What is the relationship of glucose here? survivors had a much higher glucose levels than non-survivors, and we know that hyperglycemia leads to K exit from cells too? What is the interplay of lactate and glucose on pH and K? Is the hyperK just a consequence of what is happening and the mildly elevated K is not truly "protective"
- The tables are confusing: Survivors vs non-survivors would read better than survivors vs. in-hospital mortality
- There are many table with confusing data. For example: MV disease is seen in 29% of entire group, with 23% having 2VD and 8% with 3VD. This does not add up, unless I am reading it incorrectly. This is seen on many variables.
- CRP in survivors and non-survivors is extremely different. 64% vs 5.5%. How is that not statistically significant? Its clinically significant and warrants evaluation. Data in cardiogenic shock suggest that a higher CRP on presentation is associated with mortality. This is reverse. Please elaborate
- Discussion is too long.
Author Response
Chyrchel et al present a small retrospective study on prognostic factors in patients with sudden cardiac arrest and 2 acute myocardial infarction undergoing percutaneous 3 interventions with the LUCAS-2 system for mechanical 4 cardiopulmonary resuscitation, with the main finding being those with serum potassium >5.0 mmol/L had 4.61 times higher odds of survival until discharge from the hospital. This is a retrospective study and the results should be interpreted very carefully.
- In discussing potassium levels in an arrest case, we have to emphasize and discuss pH. Ineffective CPR or delayed CPR is associated with low pH, and survival is dismal with lower pH. Why is this different here? Are the findings independent of pH? What was lactate level in each group? What is the relationship of glucose here? survivors had a much higher glucose levels than non-survivors, and we know that hyperglycemia leads to K exit from cells too? What is the interplay of lactate and glucose on pH and K? Is the hyper K just a consequence of what is happening and the mildly elevated K is not truly "protective"
This comment emphasizes some factors that are obviously of the highest importance when considering patients during SCA. These factors were among the first that we investigated during our analyses, however, we actually did not find any significant relationships that involved them (thus we did not specifically elaborate this issue in the manuscript). We investigated possible phenomena to explain the detailed mechanism of our observations.
The foundation for all effects observed in animal studies and possibly in our work is the hypothetical mechanism by which mild hyperkalemia protects the cardiac muscle at the cellular level. During cardiac arrest, circulating blood does not collect excess potassium from heart muscle cells. Tissue damage and complex processes involving intracellular potassium efflux lead to a rapid increase in potassium ions concentration in interstitial heart tissue, with a peak of 10-15 mmol/L within the first 10 minutes of ischemia. It could be even higher after reperfusion, leading to massive production of free radicals. It takes even 20–30 minutes for the gap junctions between the cardiomyocytes to close, weakening the impact of the damaged tissue on arrhythmia [Weiss 2017, doi:10.1161/CIRCEP.116.004667]. This is why we argue that systemic hyperkalemia must be considered differently during AMI with SCA – the potassium level in blood is still 2-3 times lower than in the cardiac muscle.
It seems that the protective mechanism of mild hyperkalemia in myocardial cells during SCA could stem from the adaptation of myocardial cells to higher potassium concentrations. In CKD, there are numerous ion disturbances that, by affecting the membrane potential, reduce the sensitivity of cells to interstitial hyperkalemia. Therefore, interstitial hyperkalemia appears to be less harmful as the change in potassium flux is less noticeable (as defined by the derivative of the Goldman equation). Since CPR conduced with LUCAS-2 provides up to 60% of physiological blood flow through the circulatory system [Wagner 2010, doi:10.1016/j.resuscitation.2009.11.006], it might be enough to enable ROSC in some subjects. Maintenance of this state would also be facilitated by the phenomena discussed.
As suggested, rather shortening instead of broadening the Discussion section, we did not added the above considerations to the manuscript (however, we believe that this explanation will be beneficial and enable easier understanding of the investigated phenomena). At this moment, we decided to just remark that these factors (i.e., pH and glycemia) were specifically considered (see lines 213-215).
The difference in glucose concentration between the groups is truly nonsignificant, as the distribution of the data was skewed and the non-parametric Mann-Whitney U test was performed to evaluate the difference between these two groups.
Data on lactate concentration was not recorded in a significant number of patients. Nevertheless, if such data is deemed necessary, we can collect and add all the available information from the registry in the nearest possible time.
- The tables are confusing: Survivors vs non-survivors would read better than survivors vs. in-hospital mortality
We corrected this.
- There are many table with confusing data. For example: MV disease is seen in 29% of entire group, with 23% having 2VD and 8% with 3VD. This does not add up, unless I am reading it incorrectly. This is seen on many variables.
All confusing data has been corrected, they were caused mostly due to the rounding of the numbers or due to typographic mistake in the table. These mistakes did not make any impact on the analysis
- CRP in survivors and non-survivors is extremely different. 64% vs 5.5%. How is that not statistically significant? Its clinically significant and warrants evaluation. Data in cardiogenic shock suggest that a higher CRP on presentation is associated with mortality. This is reverse. Please elaborate
We considered this important relationship during our statistical analysis. However, after performing nonparametric tests—known to be robust to outliers—it was revealed that after taking such data points into account, there was no significant difference between the groups. We have now included our explanation in the Discussion section of the manuscript to clarify this perplexing finding and eliminate possible confusion (lines 271-276).
- Discussion is too long.
As the second Reviewer requested the further elaboration of possible mechanisms that could explain observed phenomena (what would require the further extension of the Discussion section instead of shortening it), we believe that it should be left to the Editor’s decision – whether we should present more detailed considerations (incorporating the ideas that we presented in the response to Your first comment) or rather make the Discussion as brief as possible.
Reviewer 2 Report
This is a small-sample retrospective study at single center.
Accordingly, there should be several biases that could affect the endpoints, and I think that this analysis alone is insufficient to draw conclusions.
The specific association of hyperkalemia with the performance of CPR with the LUCAS-2 device needs further investigation and discussion.
In addition, I do not understand the need for Tables S1, S2 and S3. If any, authors should state the scientific rationale for that information. Instead, patient flow diagram is needed to better understand the patient selection.
Minor
Tables: the results of the categorical variables should be shown as number (%).
Author Response
This is a small-sample retrospective study at single center.
Accordingly, there should be several biases that could affect the endpoints, and I think that this analysis alone is insufficient to draw conclusions.
The proper explanation was added to the limitations (lines 329-335).
The specific association of hyperkalemia with the performance of CPR with the LUCAS-2 device needs further investigation and discussion.
As the second Reviewer requested the shortening of the discussion, we believe that it should be left to the Editor’s decision – whether we should present more detailed considerations or rather make the Discussion as brief as possible. However, we present a reasoning that we believe could be added to the discussion.
The foundation for all effects observed in animal studies and possibly in our work is the hypothetical mechanism by which mild hyperkalemia protects the cardiac muscle at the cellular level. During cardiac arrest, circulating blood does not collect excess potassium from heart muscle cells. Tissue damage and complex processes involving intracellular potassium efflux lead to a rapid increase in potassium ions concentration in interstitial heart tissue, with a peak of 10-15 mmol/L within the first 10 minutes of ischemia. It could be even higher after reperfusion, leading to massive production of free radicals. It takes even 20–30 minutes for the gap junctions between the cardiomyocytes to close, weakening the impact of the damaged tissue on arrhythmia [Weiss 2017, doi:10.1161/CIRCEP.116.004667]. This is why we argue that systemic hyperkalemia must be considered differently during AMI with SCA – the potassium level in blood is still 2-3 times lower than in the cardiac muscle.
It seems that the protective mechanism of mild hyperkalemia in myocardial cells during SCA could stem from the adaptation of myocardial cells to higher potassium concentrations. In CKD, there are numerous ion disturbances that, by affecting the membrane potential, reduce the sensitivity of cells to interstitial hyperkalemia. Therefore, interstitial hyperkalemia appears to be less harmful as the change in potassium flux is less noticeable (as defined by the derivative of the Goldman equation). Since CPR conduced with LUCAS-2 provides up to 60% of physiological blood flow through the circulatory system [Wagner 2010, doi:10.1016/j.resuscitation.2009.11.006], it might be enough to enable ROSC in some subjects. Maintenance of this state would also be facilitated by the phenomena discussed.
In addition, I do not understand the need for Tables S1, S2 and S3. If any, authors should state the scientific rationale for that information. Instead, patient flow diagram is needed to better understand the patient selection.
These additional tables have been initially added to increase the transparency of our collected data. Tables S1, S2 and S3 have now been removed from the main text and will be submitted as supplementary data. The patient flow diagram is now presented as Figure 1. We hope that these changes will suffice. If warranted, we ask for further, detailed suggestions regarding this issue.
Minor
Tables: the results of the categorical variables should be shown as number (%).
The format of the data in tables has been changed.
Round 2
Reviewer 1 Report
I appreciate changes authors made. One suggestion towards the end, that this is a "pilot study". This was a retrospective study not a pilot study.
Author Response
I appreciate changes authors made. One suggestion towards the end, that this is a "pilot study". This was a retrospective study not a pilot study.
This was modified.
Reviewer 2 Report
Authors responded to my comments.
I understand the authors' arguments.
Author Response
Thank you for your accomodating comments